# Signature Ions in MS/MS Spectra for Dansyl-Aminohexyl-QQIV Adducts on Lysine

**DOI:** 10.3390/molecules25112659

**Published:** 2020-06-08

**Authors:** Lawrence M. Schopfer, Oksana Lockridge

**Affiliations:** Eppley Institute, University of Nebraska Medical Center, Omaha, NE 68198, USA; lmschopf@unmc.edu

**Keywords:** mass spectrometry, Protein Prospector, transglutaminase, dansylQQIV, butyrylcholinesterase, plasma proteins

## Abstract

Bacterial transglutaminase was used to label human plasma proteins with fluorescent tags. Protein lysines were modified with dansyl-epsilon-aminohexyl-Gln-Gln-Ile-Val-OH (dansylQQIV), while protein glutamines were modified with dansyl cadaverine. Labeled proteins included human butyrylcholinesterase, apolipoprotein A-1, haptoglobin, haptoglobin-related protein, immunoglobulin heavy chain, and hemopexin. Tryptic peptides were analyzed by LC-MS/MS on an Orbitrap Fusion Lumos mass spectrometer. Modified residues were identified in Protein Prospector and Proteome Discoverer searches of mass spectrometry data. The MS/MS fragmentation spectra from dansylQQIV-modified peptides gave intense peaks at 475.2015, 364.1691, 347.1426, 234.0585, and 170.0965 *m*/*z*. These signature ions are useful markers for identifying modified peptides. Human butyrylcholinesterase retained full activity following modification by dansylQQIV or dansyl cadaverine.

## 1. Introduction

Protein-to-protein crosslinking reactions can be catalyzed by transglutaminase [1]. The side chain of glutamine makes a transient bond with the active site cysteine of transglutaminase accompanied by the loss of a molecule of ammonia [2], followed by reaction with the side chain of lysine to produce the epsilon-gamma-glutamyl lysine isopeptide bond. Professor Lorand developed small molecules to identify specific residues involved in transglutaminase-induced crosslinks [3,4]. In Figure 1A, the glutamine residues in a target protein are modified by transglutaminase-catalyzed incorporation of dansyl cadaverine. In Figure 1B, the lysine residues in a target protein are modified by transglutaminase-catalyzed incorporation of dansyl-aminohexyl-QQIV (dansylQQIV). These fluorescent tags were used to identify eight glutamines and 10 lysines in human Tau protein [5] susceptible to the crosslinking action of transglutaminase. Transglutaminases from human erythrocytes, guinea pig liver, lobster muscle, bovine lens, and rabbit lens, as well as fibrinoligase from human plasma, make isopeptide bonds between the side chains of glutamine and lysine [1,4,6,7,8,9]. In addition, the commercially available bacterial transglutaminase makes gamma-glutamyl-epsilon-lysine bonds [10,11,12], with the advantage that bacterial transglutaminase requires no calcium, no reducing agent, and is less costly.

Mass spectrometry has made it possible to readily identify specific residues involved in transglutaminase-catalyzed reactions. Tandem mass spectrometry fragmentation (MS/MS fragmentation) generates masses that can be used to identify the amino acid sequence of the labeled peptide. These sequence-specific masses also provide a means for identifying the modified residue. When dansyl cadaverine or biotin cadaverine is used for labeling, the MS/MS spectra of the labeled peptides include a set of prominent masses that are not specific to the amino acid sequence. These masses derive from breakdown of dansyl cadaverine and biotin cadaverine adducts on protein glutamines [10] and are characteristic indicators for the presence of the adduct. The presence of these signature ions adds confidence to the interpretation that a particular peptide is labeled by dansyl cadaverine or biotin cadaverine on glutamine. In the present report we examined MS/MS spectra for dansylQQIV adducts on protein lysines. We report the structures and masses of the signature ions that define a peptide modified by dansylQQIV on lysine.

## 2. Results

### 2.1. Plasma Proteins in the Partially Purified Human Butyrylcholinesterase Preparation

Liquid chromatography-tandem mass spectrometry (LC/MS/MS) of the tryptic digest of partially purified, native (nontagged) human butyrylcholinesterase identified butyrylcholinesterase (P06276) as the principal component in the preparation. Minor components included albumin (P02768), haptoglobin (P00738), haptoglobin-related protein (P00739), apolipoprotein A-1 (P02647), immunoglobulin (P01876), alpha-1-antitrypsin (P01009), hemopexin (P02790), and serotransferrin (P02787). Peptide counts are the number of peptides identified from a given protein. The concentration of butyrylcholinesterase in terms of peptide counts was 1114, while peptide counts for the other proteins ranged from 47 for albumin to 18 for serotransferrin.

### 2.2. Sodium Dodecyl Sulfate (SDS) Gel Electrophoresis of Proteins Labeled by DansylQQIV or Dansyl Cadaverine

Bacterial transglutaminase catalyzed the incorporation of dansylQQIV and dansyl cadaverine into native human butyrylcholinesterase and other proteins. In Figure 2A the fluorescence intensity was higher for dansylQQIV- than for dansyl-cadaverine-labeled proteins. In Figure 2B, Coomassie-blue-stained bands are equivalent for all four samples. The most intense Coomassie signals are for butyrylcholinesterase at 85 and 170 kDa, confirming the mass spectral observation that other plasma proteins are present at relatively low levels compared to butyrylcholinesterase. The fluorescence intensity for some non-butyrylcholinesterase bands in the dansylQQIV sample (Figure 2A lane 4) is as high as that for butyrylcholinesterase despite there being minimal Coomassie blue staining in these areas. The dansyl cadaverine fluorescence is most intense for butyrylcholinesterase (Figure 2A lane 3).

Though transglutaminase has the ability to crosslink proteins into high molecular weight aggregates, the control sample in Figure 2B lane 2 for native proteins treated with transglutaminase alone, shows that the experimental conditions did not produce high molecular weight protein bands.

### 2.3. Nondenaturing Gel Electrophoresis

Nondenaturing gel electrophoresis of proteins labeled by dansylQQIV or dansyl cadaverine under nondenaturing conditions resolved dansylQQIV-labeled proteins into six fluorescent bands in Figure 3A lanes 8–10, and dansyl-cadaverine-labeled proteins into at least 3 fluorescent bands Figure 3A lanes 4–6. After collecting the fluorescent image (Figure 3A), the gel was stained with Coomassie blue (Figure 3B). Some, but not all, of the fluorescent bands have a corresponding blue band. As was the case with SDS gel electrophoresis, fluorescence intensity did not correlate with Coomassie blue band intensity.

The butyrylcholinesterase activity stained gel in Figure 3C identified the location of the butyrylcholinesterase tetramer band on the nondenaturing gel. This allowed assignment of butyrylcholinesterase to the most intense blue band in Figure 3B and to the top fluorescent band in Figure 3A. The intensity of the activity stain for all three lanes that contained treated butyrylcholinesterase (lanes 12, 13, and 14) was the same as that in the untreated control (lane 11). This showed that covalent modification of butyrylcholinesterase by dansyl cadaverine and dansylQQIV had no detrimental effect on butyrylcholinesterase activity.

Though butyrylcholinesterase is the most abundant protein, the fluorescent band for dansylQQIV-labeled butyrylcholinesterase in Figure 3A lanes 8–10 is not more intense than fluorescence for other proteins. In contrast, fluorescence for dansyl-cadaverine-labeled proteins is most intense for butyrylcholinesterase.

### 2.4. Butyrylcholinesterase Activity

Dansyl-cadaverine- and dansylQQIV-modified butyrylcholinesterase samples retained full butyrylcholinesterase activity. Activities were 380 ± 12 U/mL butyrylcholinesterase control, 376 ± 10 U/mL (butyrylcholinesterase + transglutaminase), 407 ± 11 U/mL (dansyl cadaverine butyrylcholinesterase), and 367 ± 9 U/mL (dansylQQIV butyrylcholinesterase). It follows that butyrylcholinesterase can be modified by fluorescent adducts with no loss of butyrylcholinesterase activity. This observation is consistent with the activity stained gel in Figure 3C.

### 2.5. Mass Spectrometry Results for DansylQQIV Adducts on Human Butyrylcholinesterase

Transglutaminase catalyzes the formation of an isopeptide bond between the epsilon amino group of lysine and the gamma glutamyl group of glutamine. In the example shown in Figure 4, the isopeptide bond is between lysine 276 of peptide TLNLAK_276_LTGDCSR from native human butyrylcholinesterase and the second glutamine in dansylQQIV. Assignment of the crosslink to the second glutamine is discussed below.

During collision-induced dissociation in the mass spectrometer, fragment ions consistent with the amino acid sequence of this peptide (a complete y-ion sequence and a partial b-ion sequence) were observed. In addition, nonsequence ions at 170.10, 234.06, 347.14, 364.17, and 475.20 Da are present in Figure 4. These are consistent with fragments from the dansylQQIV moiety. In Figure 4, these fragments are boxed to highlight their significance. They are characteristic indicators for the presence of dansylQQIV. These boxed ions are very useful for identifying dansylQQIV-modified peptides. Their presence confirms a dansylQQIV adduct. Their absence weakens the possibility that a particular peptide is modified by dansylQQIV.

Both Protein Prospector and Proteome Discoverer found dansylQQIV adducts on butyrylcholinesterase peptides at K276 (248), K342 (314), K376 (348), K383 (355), K436 (408), K455 (427), and K556 (528) (Figure 5). Residue numbers without the 28 amino acid signal peptide are in parentheses. The MS/MS spectrum for each peptide included an added mass of 815.38 Da on lysine, substantial y-ion and b-ion sequence information, and all of the signature fragments ions of dansylQQIV. Figure 5 shows that the dansylQQIV-labeled lysines are located on the surface of the butyrylcholinesterase molecule.

### 2.6. Proposed Structures of the Signature ions Associated with Fragmentation of DansylQQIV

Figure 6 shows the structure of the parent compound, dansyl-aminohexyl-QQIV (dansylQQIV), with a neutral, monoisotopic mass of 832.4153 Da and the proposed structures of fragment ions derived from the parent ion. The 475.2015 ± 0.0004 (0.84 ppm) monoisotopic mass is singly charged and derives from fragmentation of the GlnGln linkage in dansylQQIV. It contains a heterocyclic oxazolone ring consistent with the commonly accepted oxazolone chemistry for fragmentation of peptides via collision induced dissociation in the mass spectrometer [15]. The 475.2015 ion can only be produced if the isopeptide bond to the peptide lysine is with the second Q in dansylQQIV. No 475.2015 ion could be produced if the isopeptide bond were with the first Q.

The 364.1691 ± 0.0005 (1.37 ppm) monoisotopic mass is singly charged and is consistent with dansyl aminohexyl plus NH_3_. The 347.1426 ± 0.0004 (1.15 ppm) monoisotopic mass is singly charged and can be envisioned as the product of an attack of the amino hexyl amine on the carbonyl carbon of the amino hexyl group. This reaction is conceptually similar to the aziridinone cyclization mechanism for fragmentation of peptides [16]. The 234.0585 ± 0.0001 (0.42 ppm) monoisotopic mass is singly charged and is consistent with the dansyl moiety (5-naphthalene-1-sulfonyl). This mass has been described as a fragment in the collision-induced dissociation of didansyl putrescine [17]. The 170.0965 ± 0.0001 (0.59 ppm) monoisotopic mass is singly charged and is comparable to dimethylamino naphthalene which could be formed by removal of the sulfonyl moiety from dansyl. This mass has been described as a fragment in the collision-induced dissociation of didansyl putrescine [17].

### 2.7. Second Q in DansylQQIV is the Preferred Glutamine Donor

To identify the glutamine in dansylQQIV that is the preferred substrate for bacterial transglutaminase, we searched for dansylQQ adducts (added mass of 603.23 Da) in which Ile and Val were missing and for dansylQ adducts (added mass of 492.2 Da). The search parameters for dansylQQ adducts in Protein Prospector/Batch-Tag were modified to include an elemental composition of C_28_O_8_N_6_SH_40_ and mass modification range 602 to 604 Da. We found that the same seven lysines in butyrylcholinesterase that were modified by dansylQQIV were also modified by dansylQQ. All adducts had an added mass of 603.23 Da on lysine, had substantial y-ion and b-ion sequence information, and had two to five of the signature fragment ions. Thus, these assignments are well supported. We hypothesize that dansylQQ was formed when residues Ile and Val were clipped off during trypsin digestion.

A search for dansylQ adducts with an added mass of 492.2 Da identified only one adduct, K556 of butyrylcholinesterase. Signature fragment ions were present at 170.09, 347.14, and 364.16 Da. Absence of the dansylQ signature ion at 475.20 Da is consistent with an isopeptide bond between the first glutamine of dansylQ and K556. The fact that only one positive hit had an isopeptide bond with the first glutamine in dansylQQIV supports the proposal that the second Q in dansylQQIV is the preferred glutamine donor.

Another argument that supports the second Q in dansylQQIV as the preferred glutamine donor is the structure of the 475.2015 ion. The 475.2015 *m*/*z* ion in Figure 6 represents dansylQ without residues QIV. This indicates that the first glutamine in dansylQQIV readily breaks away from the dansylQQIV adduct in the mass spectrometer, leaving the second Q linked to lysine. It follows that the first glutamine in dansylQQIV is not involved in the isopeptide crosslink. It is concluded that the second glutamine in dansylQQIV is crosslinked to the peptide lysine.

### 2.8. Mass Spectrometry Results for DansylQQIV and Dansyl QQ Adducts on Plasma Proteins Other than Butyrylcholinesterase

Database searches for plasma proteins other than butyrylcholinesterase were performed by Protein Prospector/Batch-Tag Web on Orbitrap data files. These searches were evaluated for dansylQQIV and dansylQQ adducts. DansylQQIV adducts were found on one lysine in haptoglobin, one lysine in haptoglobin-related protein, and one lysine in apolipoprotein A-1. DansylQQ adducts were found on one lysine in immunoglobulin heavy chain constant alpha 1, two lysines in hemopexin, and two lysines in apolipoprotein A-1 (Table 1). The presence of signature ions of dansylQQIV in each MS/MS spectrum supported the conclusion that the peptide was labeled on lysine by dansylQQIV or dansylQQ. No dansylQQIV adducts were found on albumin, alpha-1-antitrypsin, or serotransferrin. The protein concentration of plasma proteins in the butyrylcholinesterase preparation, other than butyrylcholinesterase, was low as indicated by Coomassie blue staining in Figure 3B. Higher protein concentrations in the labeling reaction may yield dansylQQIV adducts on most plasma proteins.

### 2.9. Dansyl Cadaverine Adducts on Glutamine

We have previously reported the signature ions associated with dansyl cadaverine adducts on glutamine to have masses of 170.0965, 234.0585, 336.1746, 402.1851, and 447.2066 using beta-casein as a model protein [10]. These masses correspond to dansyl fragments and to the immonium ions from labeled glutamine. The present report demonstrates that dansyl cadaverine can label other proteins on glutamine, namely butyrylcholinesterase and the minor plasma proteins in a partially purified butyrylcholinesterase preparation. Labeled residues and their associated proteins are listed in Table 1. The added mass from dansyl cadaverine (318.14 Da) was found for peptide adducts from each protein. The associated signature ions in MS/MS spectra produced by fragmentation of the dansyl cadaverine adducts on glutamine included the immonium ions from labeled glutamine at 447.2066, 402.1851, and 336.1746 Da plus dansyl fragmentation products at 234.0585, and 170.0965 Da.

## 3. Discussion

### 3.1. Value of Signature ions

A dansylQQIV-labeled peptide has an added mass of 815.38 Da on lysine. To find dansylQQIV adducts with Protein Prospector, we searched for adducts with an added mass in the range of 815 to 816 Da, on lysine. This can yield a large number of candidates depending on the other search parameters. We evaluated each MS/MS candidate spectrum manually. The criteria we used to validate a dansylQQIV-labeled peptide included the following. (1) At least one y-ion, b-ion, or internal fragment includes the added mass of 815.38 Da. For example, in Figure 4 the added mass of 815.38 is included in y7, y8, y9, and y10 ions. (2) The parent ion mass must include the added mass. (3) There must be signature fragment ions derived from the dansylQQIV adduct in the spectrum, namely the boxed ions in Figure 4. Signature ions were critical to the identification of dansylQQIV-labeled peptides. Search results that did not include signature ions in the MS/MS spectrum were judged to be false results.

Orbitrap data can contain dozens of MS/MS spectra for the same peptide. The Protein Prospector search engine generally selects only the best one or two from that list. Access to multiple MS/MS spectra is obtained through Xcalibur/Qual Browser software. Examination of additional spectra can reveal the presence of confirming ions that may be missing in the Protein Prospector spectrum.

For some peptides, we use an abbreviated set of criteria for identification of a dansylQQIV-labeled peptide. We require the presence of the signature ions, plus the added mass on the parent ion. Two scenarios describe when this abbreviated analysis becomes important. First, the peak heights for signature ions are frequently greater than the peak heights for fragment ions that contain the added mass of 815.38 on lysine. Therefore, an MS/MS spectrum that includes signature ions but no visible peak for a confirming ion, can be accepted as correctly representing a dansylQQIV-modified peptide, if the parent ion mass includes the added mass for dansylQQIV. Second, it is frequently not possible to fully sequence large peptides. If a large peptide contains a labeled lysine near the middle of the sequence, it may not be possible to obtain a fragment interval that contains the labeled residue. Because the signature ions have a low mass and are generally intense, they are likely to be visible if they are present. Therefore the presence of signature ions alone would be sufficient to identify a large, labeled peptide if the parent ion mass included the added mass of dansylQQIV.

### 3.2. Search Parameter Settings Make A Huge Difference

The search parameters in Protein Prospector that are critically important for getting reliable hits for dansylQQIV adducts are (1) database, (2) parent tolerance, (3) fragment tolerance, and (4) mass modification range. We got the most positive hits when we chose database SwissProt.2017, Taxonomy Homo Sapiens, and inserted the UniProt accession number of one protein at a time in the space under the heading Pre-Search Parameters. Tolerances of 20 ppm for the parent and 30 ppm for fragment ions gave reliable positive hits, but tolerances of 200 and 300 ppm did not. A mass modification range of 815 to 816 Da gave reliable hits, but a more narrow range of 815 to 815.5 Da yielded almost no hits even though the actual added mass was 815.38 Da. Though we use a high grade of trypsin, the digest included peptides cleaved at non-trypsin-specific sites. This was very helpful because a search using the term No Enzyme yielded short peptides that increased coverage and therefore increased the number of positive hits.

As is commonly reported for results from different search engines, we obtained somewhat different results from database searches with Proteome Discoverer than with Protein Prospector/Batch-Tag Web. The principal difference in search parameters between these searches was that Proteome Discoverer employed the entire SwissProt database whereas we set Batch-Tag to search a single protein at a time. The major differences in results occurred for the minor proteins in the sample, where the Proteome Discoverer search identified fewer candidates than Batch-Tag.

### 3.3. Signature Ions Do Not Derive from Free DansylQQIV

The fact that the five signature ions in Figure 6 all derive exclusively from dansylQQIV opens the possibility that these ions actually come from free dansylQQIV and not from dansylQQIV-labeled peptides. This possibility was unlikely in view of the fact that the samples were extensively dialyzed before analysis and that the labeled peptides eluted from the C18 column during LC-MS/MS at times ranging from 33.6 to 44.4 min. To test this conclusion, the free dansylQQIV molecule was subjected to LC-MS/MS under the same conditions. It eluted as a sharp peak at 42.5 min. The MS/MS spectrum showed fragments at 170.0965, 347.1426, 364.1691, and 475.2015 Da, with no indication of the 234.0585 Da mass. Taken together, these results strongly argue that the signature ion masses did not derive from free dansylQQIV.

### 3.4. More Intense Fluorescent Bands for DansylQQIV Adducts than For Dansyl Cadaverine Adducts

We have no explanation for the greater fluorescence of proteins modified by dansylQQIV compared to dansyl cadaverine. The same dansyl fluorophore is present in both adducts. Based on the number of modified residues, dansyl-cadaverine-labeled proteins might have been expected to be more fluorescent. For example, apolipoprotein A-1 had eight dansyl-cadaverine-labeled peptides, but only one dansylQQIV-labeled peptide. In general the degree of labeling of any particular residue was no greater than 5%. Comparison of peptide counts gave no evidence that the extent of labeling was higher for dansylQQIV than for dansyl cadaverine.

### 3.5. No Loss of Butyrylcholinesterase Activity for Fluorescent Adducts Produced by the Action of Bacterial Transglutaminase

Proteins that are chemically derivatized can lose enzyme activity. Human butyrylcholinesterase chemically modified with the fluorescent IRDye 800CW [18] lost 80% of its activity. In contrast, human butyrylcholinesterase enzymatically modified by bacterial transglutaminase with the fluorescent dansylQQIV or dansyl cadaverine lost no activity. Adduct formation on IgG by bacterial transglutaminase occurred with retention of antibody activity [19]. Crosslinking of soy bean peroxidase to protein G occurred with retention of both peroxidase enzyme activity and protein G binding specificity [20].

## 4. Materials and Methods

### 4.1. Materials

Dansyl-epsilon-aminohexyl-Gln-Gln-Ile-Val-OH (Zedira GmbH, product number D001, CAS number not available); a fluorescent glutamine-containing substrate for transglutaminase. Stock solution 20 mM in DMSO was stored at −80 °C. Dansyl cadaverine [*N*-(5-aminopentyl)-5-(dimethylamino)-1-naphthalenesulfonamide], BioChemika cat # 30432 CAS 10121-91-2. Stored as a 20 mM solution in dimethylsulfoxide at −20 °C. Recombinant microbial (Bacterial) transglutaminase from *Streptomyces mobaraensis* produced in *E. coli*, Zedira cat # T001 4.5 mg/mL, 139 U/mL UniProt Accession number P81453. Human plasma proteins in partially purified human butyrylcholinesterase [21] UniProt Accession number P06276. Trypsin from porcine pancreas, Sigma cat # T6667 lyophilized 20 µg per vial. Precast 4–20% gradient polyacrylamide gels, BioRad cat# 456-8084. Slide-A-Lyzer 7K dialysis cassettes, Pierce No. 66373.

### 4.2. Transglutaminase-Catalyzed Incorporation of DansylQQIV and Dansyl Cadaverine into Human Plasma Proteins

A side fraction of human butyrylcholinesterase (P06276) partially purified from human plasma was the source of a mixture of plasma proteins. The protein mixture was diluted to 1 mg/mL in 20 mM imidazole chloride pH 7.5 for reaction with 0.4 mM dansylQQIV or 0.4 mM dansyl cadaverine. The reaction was catalyzed by 0.5 U/mL bacterial transglutaminase in a total volume of 0.5 mL. Control samples with or without transglutaminase were also prepared. Samples were incubated at 37 °C in a humidified chamber for 14 h.

### 4.3. Butyrylcholinesterase Activity Assay

Samples were diluted 100 fold into 1 mg/mL bovine albumin dissolved in phosphate buffered saline. Aliquots of 20 µL were assayed in a 2 mL reaction volume containing 1 mM butyrylthiocholine iodide and 0.5 mM dithiobisnitrobenzoic acid in 0.1 M potassium phosphate buffer pH 7.0 at 25 °C. The increase in absorbance at 412 nm was recorded for 40 s. The slope was converted to units of activity using an extinction coefficient of 13.6 mM^−1^ cm^−1^. Units of activity were defined as micromoles per minute.

### 4.4. Fluorescence of DansylQQIV and Dansyl-cadaverine-labeled Proteins on Polyacrylamide Gels

Precast 4–20% acrylamide gels were electrophoresed in nondenaturing buffer (21 mM Tris, 23 mM glycine pH 9) at 200 volts constant voltage for 3 h at room temperature, or in SDS buffer for 35 min. Gels were placed in water for acquisition of fluorescent images on an Azure c600 system UV302 for 15 s. Following acquisition of a fluorescent image, gels were stained with Coomassie blue. Nondenaturing gels were stained for butyrylcholinesterase activity as described [22].

### 4.5. Sample Preparation for Liquid Chromatography Tandem Mass Spectrometry

A portion of each sample was set aside for activity assays and nondenaturing gel electrophoresis. Another portion was denatured in a boiling water bath for 3 min in the presence of 10 mM dithiothreitol. After a brief cooling period, iodoacetamide was added to a final concentration of 50 mM. The carbamidomethylated samples were injected into Slide-A-Lyzer cassettes and dialyzed against 3 × 4 L of 20 mM ammonium bicarbonate pH 8 at 4 °C to remove excess dansylQQIV, dithiothreitol, and iodoacetamide. Trypsin digestion was more complete when samples were free of excess reagents.

The protein concentration of the dialyzed samples was estimated from absorbance at 280 nm taken with a NanoDrop spectrophotometer (ND-100; NanoDrop Technologies). The protein concentration was adjusted to 4 µg/µL by reducing the volume in a vacuum centrifuge. Trypsin was added to give a protein µg to trypsin µg ratio of 20:1. Samples were incubated at 37 °C in a humidified chamber for 17 h. Trypsin was inactivated by adding formic acid to 0.1%. Samples were centrifuged for 20 min at top speed in a microfuge before transferring 50 µL to autosampler vials.

### 4.6. Liquid Chromatography Tandem Mass Spectrometry (LC-MS/MS) on the Orbitrap Fusion Lumos Mass Spectrometer

Trypsin-digested samples were analyzed on the Orbitrap Fusion Lumos mass spectrometer (Thermo Fisher), a high resolution instrument. A Thermo RSLC Ultimate 3000 ultra-high pressure liquid chromatography system (Thermo Scientific) was coupled to the Orbitrap Fusion Lumos via an Acclaim PepMap 100 C18 trap column (75 µm × 2 cm, Thermo Scientific) and a Thermo Easy-Spray PepMap RSLC C18 separation column (75 µm × 50 cm, Thermo Scientific). A 2 µL sample (about 8 µg peptides) was loaded onto the trap column, washed with 100% solvent A (0.1% formic acid in water) for 10 min at 2 µL/min, shuttled onto the separation column, and eluted with a biphasic, linear gradient of 5 to 50% solvent B (0.1% formic acid in 80% acetonitrile) over 30 min followed by 50 to 100% solvent B over 40 min, at a flow rate of 0.3 µL/min. Parent ion mass spectra were collected in the Orbitrap detector (resolution of 120,000), in positive ion mode, with a charge state of 2–6, over a mass range of 350 to 1800 *m*/*z*. Mass tolerance was 10 ppm, data collection for a given mass was excluded after the first acquisition for 30 s. Maximum injection time was 100 msec, ion transfer tube temperature was 275 °C, ion spray voltage was 1900 volts, and automatic gain control was 400,000. Fragment ion spectra were taken using data dependent acquisition, isolation was in the quadrupole, and the detector was the Orbitrap (resolution 30,000). Fragmentation was by high energy collision-induced fragmentation at 35% normalized collision energy, maximum injection time was 60 msec, automatic gain control was 50,000, and the scan range was auto normal.

### 4.7. Protein Prospector Search for DansylQQIV and Dansyl Cadaverine Modified Peptides

Mass spectrometry *.raw files were converted to *.mgf files using MSConvert v 3.0 from Proteo Wizard [23]. Data files were loaded onto Batch-Tag Web in Protein Prospector http://prospector.ucsf.edu/prospector/mshome.htm. To identify peptides labeled by dansylQQIV, Batch-Tag Web search parameters were (1) Database: SwissProt.2017.11.01, (2) Taxonomy: Homo Sapiens, (3) Pre-Search Parameters: Accession number for the protein being searched, (4) Digest: Trypsin, (5) Max. Missed Cleavages: 2, (6) Constant Mods: carbamidomethyl (C), (7) Variable Mods: Oxidation (M), (8) Parent Tol: 20 ppm, (9) Frag Tol: 30 ppm. It was important to change the default tolerances of 200 and 300 ppm to 20 and 30 ppm because the default tolerances yielded no useful results. (10) User Defined Variable Modifications: Mod 1 Label dansylQQIV; Specificity K; Mod 1 Elem Comp C39H57N7O10S, (11) Mass Modifications: range Da 815–816. The range setting was critical to getting a good result. A setting of 815–815.5 yielded almost no adducts. (12) Check mark in the K box. All other settings were left at their default values. 

To identify peptides labeled with dansyl cadaverine, Batch-Tag Web search parameters were the same as those used for dansylQQIV-labeled peptides except (1) User Defined Variable Modification: Mod 1 Label dansyl cadaverine; Specificity Q; Mod 1 Elem Comp C17H22N2O2S, (2) Mass Modifications: range Da 318-320, (3) Check mark in the Q box. MS/MS spectra for modified peptides identified by Batch-Tag Web were evaluated with Search Compare in Protein Prospector.

### 4.8. Proteome Discoverer Search for DansylQQIV Modified Peptides

Mass spectrometry data files *.raw were searched against a SwissProt database using Proteome Discoverer v 2.2 (Thermo Fisher Scientific). Search parameters consisted of (1) Processing Method: PWF Fusion Basic Sequest HT, (2) Protein Database: 20191107 Swiss Prot Homo sapiens, (3) Enzyme: trypsin (full), (4) Missed Cleavages: 2, (5) Minimum Peptide Length: 4, (6) Precursor Mass Tolerance: 20 ppm, (7) Fragment Mass Tolerance: 0.03 Da, (8) Static Modifications: carbamidomethyl (C), (9) Dynamic Modifications: oxidized (M), dansyl cadaverine (Q) monoisotopic added mass 318.1402 Da [Mo], and dansylQQIV (K) monoisotopic added mass 815.3888 Da [Mo], (10) Consensus Method: CWF Basic. All other parameters were left at default values. Exact mass values for dansyl cadaverine and dansylQQIV were calculated from their elemental compositions using Scientific Instrument Services Exact Mass Calculator (https://www.sisweb.com/referenc/tools/exactmass.htm). The MS/MS spectra for modified peptides identified by Proteome Discoverer were evaluated with Xcalibur/Qual Browser (Thermo Scientific).

## 5. Conclusions

Bacterial transglutaminase decorates the surface of proteins with fluorescent tags, with no loss of function of the tagged protein. The protein example in the present work, human butyrylcholinesterase, adds to literature reports for IgG heavy chain constant region [19], soy bean peroxidase, and protein G [20], which retained function after being modified by bacterial transglutaminase. Crosslinked proteins involved in blood coagulation, celiac disease, cataracts, and neurodegenerative diseases can be investigated with the aid of transglutaminase-catalyzed incorporation of dansylQQIV and dansyl cadaverine. The modified peptides can be identified by mass spectrometry with the aid of the signature ions we describe in this report. 

## Figures and Tables

**Figure 1 molecules-25-02659-f001:**
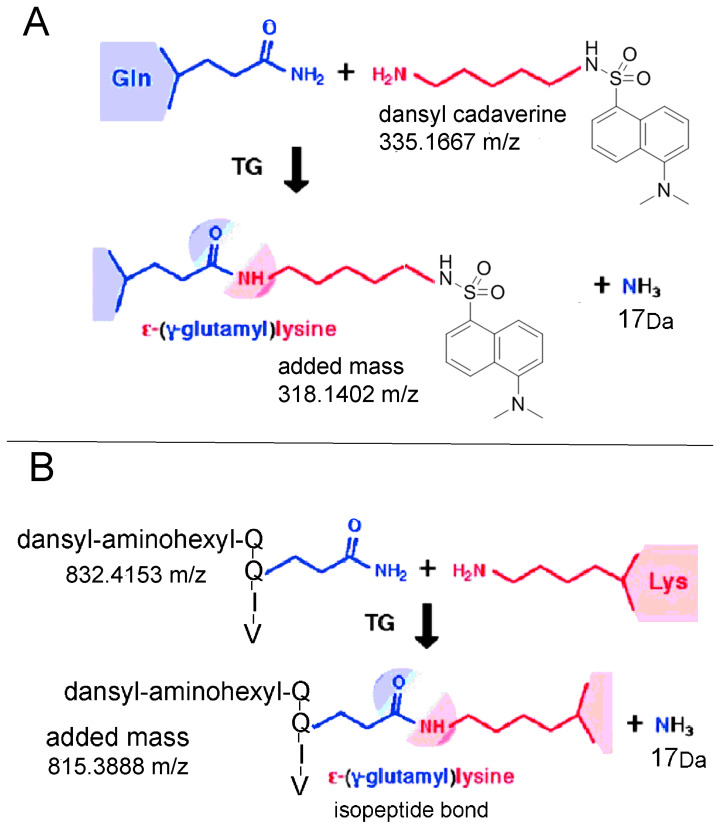
Transglutaminase (TG) catalyzes the incorporation of small molecules into target proteins. (**A**) Dansyl cadaverine makes a covalent bond with the side chain of glutamine, adding 318.1402 Da to each modified glutamine. (**B**) dansyl-aminohexyl-QQIV (dansylQQIV) makes a covalent bond with the side chain of lysine, adding 815.3888 Da to each modified lysine. Formation of the epsilon-gamma-glutamyl lysine isopeptide bond is accompanied by loss of ammonia. The color scheme was adapted from [13].

**Figure 2 molecules-25-02659-f002:**
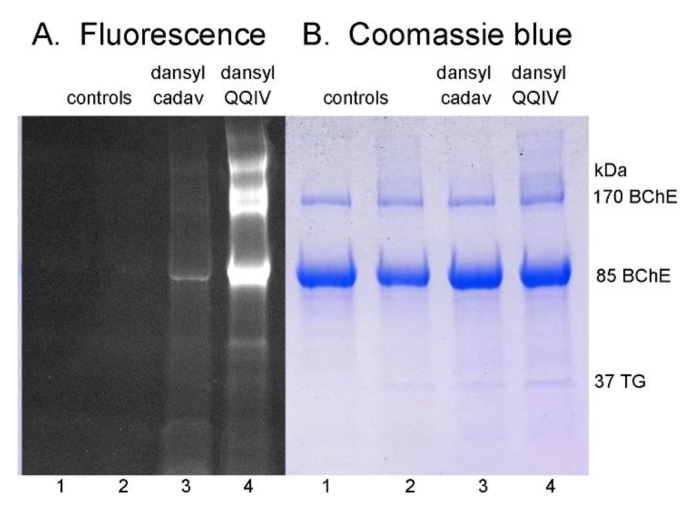
SDS gel electrophoresis of partially purified butyrylcholinesterase. Protein was loaded at 10 µg per lane. Samples for lanes 2, 3, and 4 contained transglutaminase. The control sample in lane 2 contains no dansyl label. The control sample in lane 1 contains neither transglutaminase nor the dansyl label. The fluorescent image in panel (**A**) was acquired from a 15 s exposure to UV light at 302 nm, in the Azure c600 imaging system. The fluorescent image is brighter for proteins labeled with dansylQQIV (lane 4) than for those labeled with dansyl cadaverine (lane 3). The same gel was stained with Coomassie blue in panel (**B**). Blue bands at 85 and 170 kDa support the mass spectrometry finding that butyrylcholinesterase (BChE) is the most abundant protein in the preparation. Bacterial transglutaminase (TG) in lanes 2, 3, and 4 is indicated by a faint band at 37 kDa. Fluorescence intensity for dansylQQIV-labeled proteins does not correlate with Coomassie blue intensity.

**Figure 3 molecules-25-02659-f003:**
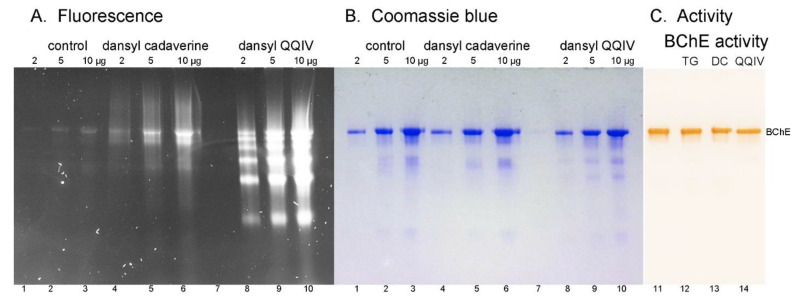
Nondenaturing gel electrophoresis of partially purified butyrylcholinesterase. Lanes are numbered 1–10 in panels (**A**,**B**) to emphasize that the same gel was imaged for fluorescence in panel (**A**) and stained with Coomassie blue in panel (**B**). Protein quantities in panels (**A**, **B**) were 2, 5, and 10 µg for the control sample (lanes 1–3), dansyl-cadaverine-labeled sample (lanes 4–6), and dansylQQIV-labeled sample (lanes 8–10). Transglutaminase-catalyzed incorporation of dansylQQIV yielded six fluorescent bands (lanes 8–10), which were more intense than the three fluorescent bands for dansyl cadaverine (lanes 4–6). The gel in panel (**C**) was stained for butyrylcholinesterase (BChE) activity (lanes 11–14). This identified the location of the butyrylcholinesterase tetramer and showed that covalent modification of butyrylcholinesterase by dansyl cadaverine (DC) and dansylQQIV had no detrimental effect on butyrylcholinesterase activity. Protein quantity in panel C was 0.02 µg (0.01 units) per lane.

**Figure 4 molecules-25-02659-f004:**
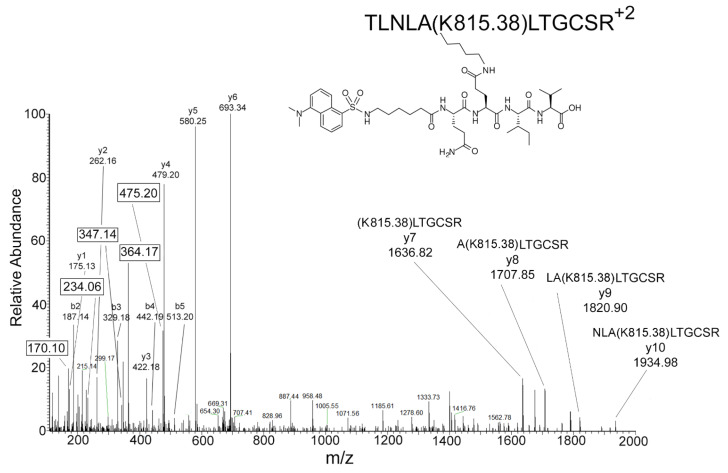
MS/MS spectrum of the dansylQQIV adduct on lysine 276 of human butyrylcholinesterase (P06276). Covalent attachment of dansylQQIV to the epsilon amino group of lysine increases the mass of the modified peptide by 815.38 Da. A complete y1 to y10 series supports the peptide structure and the modification on K. The y7, y8, y9, and y10 ions include the dansylQQIV-modified lysine. Boxed ions at 170.10, 234.06, 347.14, 364.17, and 475.20 *m*/*z* are fragments of dansylQQIV. The isopeptide bond is with the second glutamine in dansylQQIV as illustrated in the structure.

**Figure 5 molecules-25-02659-f005:**
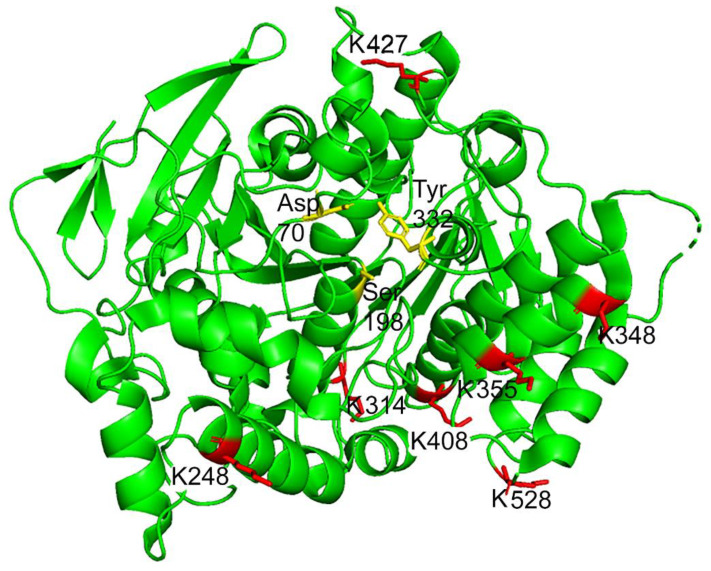
X-ray structure of the butyrylcholinesterase monomer (PDB 1P0I) annotated with the dansylQQIV-labeled lysine residues K276 (248), K342 (314), K376 (348), K383 (355), K436 (408), K455 (427), K556 (528) shown in red. Residue numbers without the 28 amino acid signal peptide are in parentheses. The yellow residues indicate the position of the active-site gorge, with Asp70 and Tyr332 at the mouth of the gorge and Ser198 of the catalytic triad at the bottom. The figure was made with PyMol [14] http://www.pymol.org.

**Figure 6 molecules-25-02659-f006:**
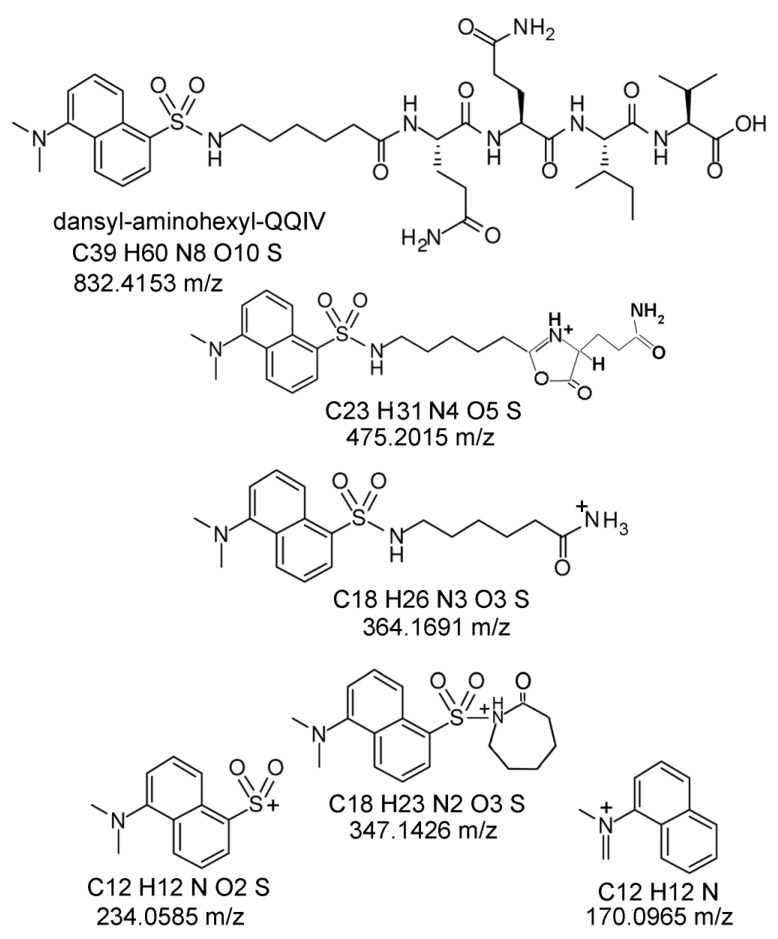
Structure of dansyl-aminohexyl-QQIV (Zedira GmbH, product number D001) and proposed structures for the five signature ions produced by collision-induced dissociation of dansylQQIV peptide adducts in the mass spectrometer.

**Table 1 molecules-25-02659-t001:** Human plasma proteins labeled by dansylQQIV or dansylQQ on lysine (K) or by dansyl cadaverine on glutamine (Q).

Protein	Accession #	DansylQQIV Lysine	Dansyl QQ Lysine	Dansyl Cadaverine Glutamine
Butyrylcholinesterase	P06276	K276, K342, K376, K383, K436, K455, K556	K276, K342, K376, K383, K436, K455, K556	Q75, Q200, Q204, Q251, Q344, Q379, Q408, Q588
Albumin	P02768	None	None	Q614
Haptoglobin	P00738	K291	None	Q400
Immunoglobulin heavy constant alpha 1	P01876	K200	K212	Q266, Q283, Q287
Alpha-1-antitrypsin	P01009	None	None	None
Hemopexin	P02790	None	K91, K466	Q133, Q155, Q390
Haptoglobin-related protein	P00739	K233	None	None
Apolipoprotein A-1	P02647	K120	K47, K131	Q56, Q87, Q93, Q129, Q141, Q151, Q162, Q240
Serotransferrin	P02787	None	None	None

Residue numbers include the signal peptide leader sequence.

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
