# Peer review of "Signature Ions in MS/MS Spectra for Dansyl-Aminohexyl-QQIV Adducts on Lysine"

_molecules, 2020, doi:10.3390/molecules25112659_

Round 1
Reviewer 1 Report
General comment:
The manuscript is a well organized, well presented and a convincing paper. The study extended the possibilities to identify proteins based on proteomic analysis involving pre-hydrolysis modification of these proteins with feasibly identifiable tags. Description of the chemistry behind the chemical derivatization as well as mass spectrometric analyses is informative, examples of practical use are included.
Minor comments:
- First few sentences of introduction may be misleading. Better to start with transglutaminase-catalyzed crosslinks and the organophosphorus crossling to be mentioned afterwards.
- Results, section 2.1. Please mention explicitely that the analysis was performed on native (i.e., non-tagged) proteins.
- Results, line 62. I am not sure what the term „peptide counts“ means? Is it a number of peptides detected?
- Results, 2.5. Formula of the peptide containing dansylQQIV lysine: instead of the form TLNLAK(815.38)LTGCSR+2 please consider using the formula TLNLA(K+815.38)LTGCSR+2.
Author Response
Please see the attached file containing our reply to reviewers' comments.

Reviewer 2 Report
The manuscript entitled “Signature ions in MS/MS spectra fordansyl-aminohexyl-QQIV adducts on lysine authored by Lawrence M. Schopfer and Oksana Lockridge describes the MS characterization of peptides modified with dansyl-epsilon-aminohexyl-Gln-Gln-Ile-Val-OH (dansylQQIV) at lysine residues.
It was not very clear the importance of characterizing lysine residues covalently modified with dansyl-epsilon-aminohexyl-Gln-Gln-Ile-Val-OH (dansylQQIV). Actually, the introduction is very difficult to follow. It was necessary to read the previous papers of the team to understand the type of modifications referred in the text. Therefore, major revisions have to be performed in this paper in order to be acceptable for publication:
- To better follow the introduction, please introduce a generic picture of the cross-links induced by Organophosphorus and of the dansyl-epsilon-aminohexyl-Gln-Gln-Ile-Val-OH (dansyl QQIV) incorporation
- The analyses were performed in a high resolution mass spectrometer, which is a great advantage to support the structural assignments of fragments. However, the authors present the m/z values with only 1 decimal place. Please provide the m/z values of the signature fragments with 4 decimal places and calculate the associated error in ppm, in order to have an idea of the validity of the assigned structures to each fragment ion.
- Please include a Conclusion section, highlighting the importance of the results.
Round 2
Reviewer 2 Report
The authors conveniently addressed/corrected all issues raised during the revision process. The paper can now be accepted in the present form.